# Rehearsal NeRF: Disentangling Dynamic Illuminations in Neural Radiance Fields

## Abstract

Although there has been significant progress in neural radiance fields, an issue on dynamic illumination changes still remains unsolved. Different from relevant works that parameterize time-variant/-invariant components in scenes, subjects' radiance is highly entangled with their own emitted radiance and lighting colors in spatio-temporal domain. In this paper, we present a new effective method to render and reconstruct neural fields under severe illumination changes, named *ReHeaRF*. Our key idea is to leverage scenes captured under stable lighting like rehearsal stages, easily taken before dynamic illumination occurs, to enforce geometric consistency between the different lighting conditions. In particular, ReHeaRF uses a learnable vector for lighting effects which represents illumination colors in a temporal dimension and is used to disentangle projected light colors from scene radiance. Furthermore, our ReHeaRF is also able to reconstruct the neural fields of dynamic objects by using off-the-shelf interactive masks for key frames. To decouple the dynamic objects, we propose a new regularizer, removing dynamic parts with similar colors to the light sources. We demonstrate the effectiveness of ReHeaRF by showing robust performances on view synthesis under dynamic illumination conditions and outperforming state-of-the-art approaches in both quantitative and qualitative evaluations. We submit our source codes and video demo as supplementary materials.

## 1 Introduction

Neural Radiance Fields (NeRFs) Mildenhall et al. (2020) represent a scene as neural implicit functions and enable to render photo-realistic images from arbitrary viewpoints. For wider applicability of NeRFs, its variant representations for dynamic motions Wu et al. (2022); Zhang et al. (2023); Park et al. (2021a;b); Li et al. (2021); Pumarola et al. (2021); Weng et al. (2022); Peng et al. (2021) have been actively studied. Existing dynamic radiance fields synthesize sequential frames with novel viewpoints by decoupling static and dynamic objects Wu et al. (2022); Zhang et al. (2023), topological deformation Park et al. (2021a;b); Li et al. (2021); Pumarola et al. (2021), and human movements Weng et al. (2022); Peng et al. (2021). It is worth nothing that the word, 'dynamic', refers subjects' motions only. In this work, we extend the definition of the 'dynamic' to varying illuminations as well as subjects' motions during taking an input video.

Estimating and manipulating scene illuminations, such as intrinsic image decomposition Barrow et al. (1978); Horn (1974), relighting Debevec et al. (2000); Xu et al. (2018) and shape-from-shading Zhang et al. (1999), have been considered as one of classical research issues. There is a common assumption in these works that light sources are stable and predictable. Nevertheless, their main challenge is that the solution is not unique due to limited image resolution, noise and inaccurate camera geometry. To alleviate the challenges, prior information, such as depth geometry Chen & Koltun (2013); El Helou et al. (2021); Maier et al. (2017), lighting directions Somanath & Kurz (2021); Wang et al. (2021b) and segmentation masks Munkberg et al. (2022), is available, making their inferences more tractable. Despite the significant efforts of the previous works, there has been no attempt to explicitly handle varying illuminations in NeRFs.

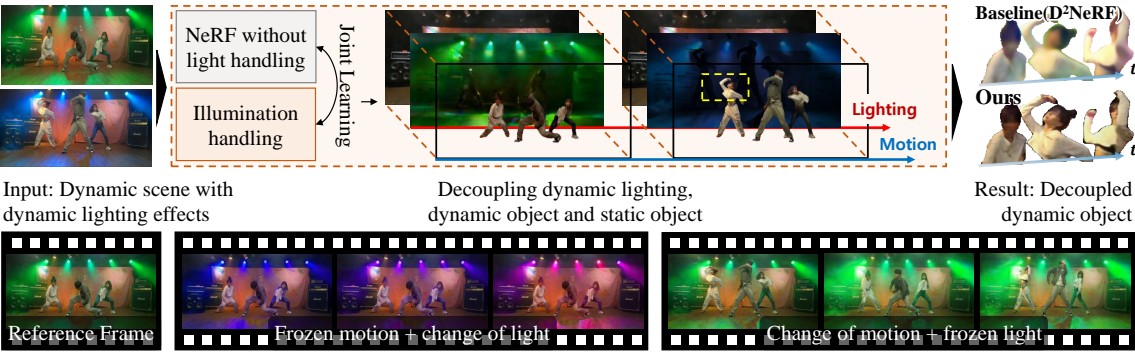

Figure 1: ReHeaRF jointly optimizes three neural fields for dynamic lighting and static/moving subjects in a training step. Each field cannot only be rendered independently, but it can also be composited at once to represent a whole scene. Compared to the baseline Wu et al. (2022), the details from our ReHeaRF are distinguishable and the rendered colors are well decoupled from the scene lights. We provide a variety of applications for video editing, such as controlling lighting while stopping motion and vice versa.

As the first step toward addressing this issue, we need to find a proper prior to reduce an ambiguity of dynamic illuminations in neural fields. Our key observation is that the dynamic illuminations, intense and rapid lighting intensity and color changes during recording input videos, are created artificially in controlled situations, such as plays and concerts. As they often come with a rehearsal stage without the dynamic lighting effects before starting the main stages, we use the video on the rehearsal stage as a prior. Although motions of the dynamic object in the rehearsal stage are not perfectly aligned with that of the main stage, it is valuable to represent scene geometry and to disentangle objects' own colors and the dynamic lighting effects in neural fields.

In this work, we present a **ReHea**rsal prior-based **R**adiance **F**ields (ReHeaRF) to synthesize novel viewpoint images of a scene under dynamic lighting environments. As shown in Fig. 1, our ReHeaRF is designed to decouple dynamically changing illuminations and dynamic/static objects, simultaneously. To do this, we exploit the fact that scene geometry is consistent regardless of the lighting changes, except for regions of moving subjects. Our ReHeaRF first takes both rehearsal and main stage videos as input, and then reconstructs three neural fields for static, dynamic objects, and dynamic illuminations from lighting effects. With our ReHeaRF, we can easily implement applications to video editing including motion and lighting controls.

To be specific, we develop a joint optimization for decoupling dynamic illuminations from moving/static objects in the scene. Without optimizing the dynamic illuminations, decoupling dynamic and static objects will not be done because the illumination across the scene varies in their neural fields. We thus introduce a learnable illumination vector to represent time-variant global and local illuminations of scenes. With a neural field for illuminations, the vector is used for rendering the dynamically changing illuminations over time. To jointly optimize the neural fields for illuminations and dynamic objects, we then propose a regularization using the learned illumination vector to enforce the disentanglement between the neural fields. With the rehearsal prior, we design an additional regularization for robust supervision to decouple static objects and illuminations. Dynamic objects belonging to the rehearsal stage may hinder the optimization of the neural field for the static objects. Our regularization alleviates the issue by pushing the decoupling process. We demonstrate the effectiveness of ReHeaRF by qualitatively and quantitatively evaluating it over state-of-the-art methods for NeRFs in the dynamic domain, even with a neural network for light effect suppression.

## 2 RELATED WORKS

**Dynamic neural scene representations** There are three types of works categorizing dynamic neural scene representations: deformation of non-rigid parts, motions with human priors and decoupling dynamic part.

To represent moving/deformable objects in a scene, D-NeRF Pumarola et al. (2021) proposes neural implicit representations consisting of deformation fields and canonical fields to handle non-rigid motions. Nerfies Park et al. (2021a) uses per-frame deformation latent codes, instead of time-stamps. HyperNeRF Park et al. (2021b) extends the Nerfies to account for topological changes using high-dimensional hyperspace. For human motions, H-NeRF Xu et al. (2021) and HumanNeRF Weng et al. (2022) use human body templates Alldieck et al. (2021); Loper et al. (2015) to train the person in a canonical space. RigNeRF Athar et al. (2022) focuses on the deformation of head-pose and facial expressions using 3D morphable face model Blanz & Vetter (1999). Here, the template misalignment often has negative impacts on the reconstruction quality. To separate moving foregrounds from static backgrounds, several works have used priors such as optical flow Li et al. (2021); Gao et al. (2021) and 3D depth Li et al. (2021); Gao et al. (2021); Xian et al. (2021) and segmentation masks Zhang et al. (2021); Tschernezki et al. (2021); Li et al. (2023); Jiang et al. (2022). Without any prior information, D²NeRF Wu et al. (2022) decouples static and dynamic components and handles shadow effects of moving objects in a self-supervised manner. NeRFPlayer Song et al. (2023) predicts the sample-wise probabilities of being static, deforming, and newness in a 4D spaciotemporal space.

However, the previous works assume that a dynamic scene is under stable illumination conditions. Open4D Bansal et al. (2020) captures sequences in a variety of environments and lighting conditions, such as performance dances including ballet and theatrical reenactments, but it remains an issue on lighting and shadows in 4D spatio-temporal representations as future research.

**Neural representations for lighting** Thanks to the powerful 3D scene and light modeling capacities of neural implicit representations, there have been widely studied in intrinsic image decomposition Munkberg et al. (2022); Boss et al. (2021b;a; 2022), relighting Srinivasan et al. (2021); Lyu et al. (2022), and shape-from-shading Ling et al. (2023); Yang et al. (2022). Works in Munkberg et al. (2022); Boss et al. (2021b;a; 2022) jointly optimize lighting, materials, and scene geometry from multi-view images using neural representations. Relighting Srinivasan et al. (2021); Lyu et al. (2022) with neural representations deals with global illuminations, 3D geometry, and material information from a set of images with unconstrained known lighting locations. NeRV Srinivasan et al. (2021) takes multiple images captured under known lighting conditions and produces a 3D representation of a scene, enabling to render of novel viewpoint images with arbitrary lighting directions. ShadowNeuS Ling et al. (2023) uses a shadow ray supervision to reconstruct neural signed distance fields from single-view images under multiple lighting conditions.

**Appearance embedding for NeRF** Following generative latent optimization Bojanowski et al. (2018), many NeRF extensions use trainable latent codes for per-image appearance variations Martin-Brualla et al. (2021); Turki et al. (2022); Tancik et al. (2022), time-varying components Li et al. (2022b); Park et al. (2021a;b) and manipulation Schwarz et al. (2020); Wang et al. (2022); Niemeyer & Geiger (2021). NeRF-in-the-wild Martin-Brualla et al. (2021) proposes a latent appearance modeling to address the scene inconsistency caused by different lighting conditions of unstructured photo-collections. Mega-NeRF Turki et al. (2022) and Block-NeRF Tancik et al. (2022) handle the issue in large-scale scenes by embedding scene appearances like illumination changes according to camera poses into latent codes. DyNeRF Li et al. (2022b) uses NeRF framework as a baseline and use temporal latent codes, allowing neural fields to be changed in time domain. CLIP-NeRF Wang et al. (2022) shows editable NeRF with text prompts to manipulate shapes and appearances of neural fields. GRAF Schwarz et al. (2020) and GIRAFFE Niemeyer & Geiger (2021) present 3D-aware controllable image synthesis using conditional neural fields.

## 3 METHODOLOGY

Given $N$ sampled points of a camera ray $\mathbf{r}$, NeRFs render view dependent colors $\tilde{\mathbf{C}}$ as following the volume rendering formula:

$$\tilde{\mathbf{C}}(\mathbf{r}) = \sum_{i=1}^{N} T_i \alpha_i \mathbf{c_i} \quad \text{s.t.} \quad T_i = \sum_{j=1}^{i-1} \exp(-\sigma_j \delta_j), \; \alpha_i = 1 - \exp(-\sigma_i \delta_i) \quad (1)$$

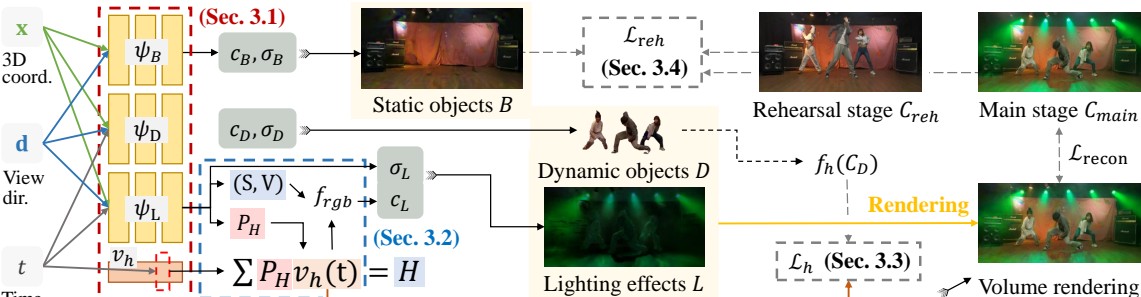

Figure 2: An overview of ReHeaRF in training phase. Our ReHeaRF consists of three neural fields whose outputs are static, dynamic objects and illuminations. Each field takes location ($\mathbf{x}$), viewing direction ($\mathbf{d}$) and time-stamp ($\mathbf{t}$) for the same sample point as input, except the time-stamp for the static fields. They are used to infer the radiance and density for each. Particularly, the illumination field predicts the probability $P_H$ of the illumination vector $v_h$ corresponding to $\mathbf{t}$. To decouple static/dynamic objects and illumination components of the main stage video well, two additional regularizations using the rehearsal prior and $v_h$ are designed.

where $\mathbf{c}_i$ and $\sigma_i$ denote the spatially-view-dependent radiance and the spatially-dependent density, respectively, which are calculated by the radiance field for the $i$-th sample. $\delta$ refers a distance between the $i$-th sample and the $i + 1$-th sample.

With the radiance fields of the scene built, we propose a novel neural scene representation pipeline which decouples components of dynamic lighting effects and objects, depicted in Fig. 2. We first handle dynamic lighting effects through separate neural fields (Sec. 3.1) with a learnable illumination vector (Sec. 3.2). Then, we jointly optimize global and local illumination on dynamic (Sec. 3.3) and static objects (Sec. 3.4) with a regularization using the illumination vector and a rehearsal prior, respectively.

## 3.1 NeRF for Dynamic Lighting Effects

Our method learns each radiance field for static, dynamic objects and illuminations caused by lighting effects, as shown in Fig. 2. For decoupling static and dynamic objects, we adopt D$^2$NeRF Wu et al. (2022) as a baseline, and compute a density $\sigma_B$ and a radiance $\mathbf{c}_B$ of a static objects $B$, and a density $\sigma_D$ and radiance $\mathbf{c}_D$ of dynamic objects $D$, to build neural scene representation as below:

$$\psi_B : (\mathbf{x}, \mathbf{d}) \rightarrow (\sigma_B, \mathbf{c}_B), \;\; \psi_D : (\mathbf{x}, \mathbf{d}, t) \rightarrow (\sigma_D, \mathbf{c}_D), \tag{2}$$

where $\mathbf{x} \in R^3$ is a 3D spatial coordinate, $t \in R$ is a temporal coordinate and $\mathbf{d} \in R^3$ is a viewing direction.

To separate the illumination from the main stage video, our ReHeaRF represents lighting effects $L$, which are colors formed by light sources. Here, we introduce a learnable illumination vector $v_h$ for a self-supervision of the light illuminations, and a illumination neural field $\psi_L$ to render time-varying lighting effects. Details for the illumination vector and rendering the lighting effects with $\psi_L$ will be described in Sec. 3.2.

We also re-formulate a composite rendering of the baseline Wu et al. (2022) which assumes that placing objects in the same position is physically impossible. However, the lighting effects exist on the surface of objects in our problem setup. Considering this property, we formulate a modified version of the composite rendering by multiplying the normalized transparency map $\rho$ as below:

$$\tilde{\mathbf{C}}_{main}(\mathbf{r}, t) = \sum_{i=1}^{N} \sum_{e} T_{main,i} \rho_{e,i} \alpha_{e,i} \mathbf{c}_{e,i} \tag{3}$$

$$\text{s.t.} \;\; T_{main,i} = \exp\left(-\sum_{j=1}^{i-1} \sum_{e} \sigma_{e,j} \delta_j\right), \;\; \rho_{e,i} = \frac{\alpha_{e,i}}{\sum_{e} \alpha_{e,i}} \;\; \text{and} \;\; \alpha_{e,i} = 1 - \exp(\sigma_{e,i} \delta_i), \tag{4}$$

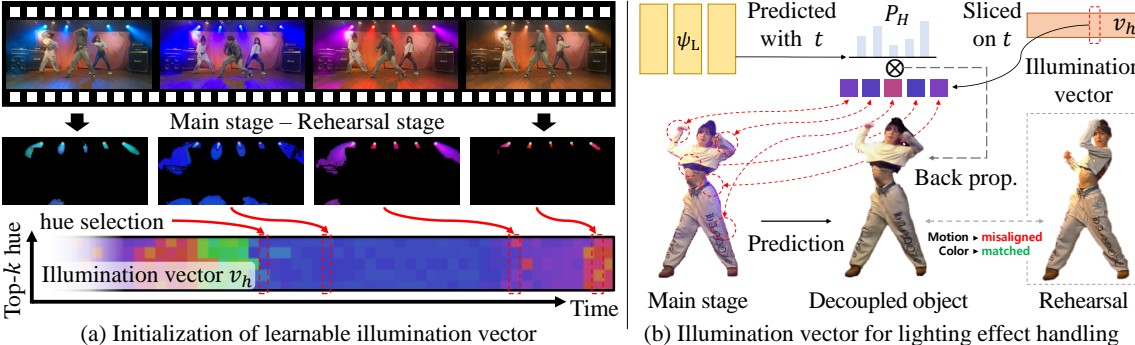

(a) Initialization of learnable illumination vector     (b) Illumination vector for lighting effect handling

Figure 3: Visualizations of the procedure how to work the illumination vector. (a) The initialization of the learnable illumination vector is done by sampling the hue channel of the dynamic lighting from the difference between the rehearsal and the main stage video. (b) Back propagation from the rendered hue channel is achieved by computing a weighted sum of the probability $P_H$ and the slice of the $v_h$ for time $t$, which finally optimize the hue channel of the dynamic lighting effects.

where $\tilde{\mathbf{C}}_{main}$ is the rendered color of the main stage, $e \in [B, D, L]$ is a type of components in the main stage, $\sigma_{e,i}$ and $\mathbf{c}_{e,i}$ denote the density and radiance of a component $e$ at $i$-th sampled point, respectively. In this equation, $\rho$ plays a role in a normalization term to avoid the color saturation problem by considering the relative weights of each component in the main stage.

Based on Eq. (1), the colors of the static background $B$, moving objects $D$ and lighting effects $L$ can be rendered individually as follows:

$$\tilde{\mathbf{C}}_L(\mathbf{r}, t) = \tilde{\mathbf{C}}_{main}(\mathbf{r}, t) - \tilde{\mathbf{C}}_B(\mathbf{r}), \;\; \text{s.t.} \;\; \tilde{\mathbf{C}}_B(\mathbf{r}) = \sum_{i=1}^{N} T_{B,i} \alpha_{B,i} \mathbf{c}_{B,i}, \tag{5}$$

$$\text{and} \quad \tilde{\mathbf{C}}_D(\mathbf{r}, t) = \sum_{i=1}^{N} T_{D,i} \alpha_{D,i} \mathbf{c}_{D,i}, \tag{6}$$

where $T_{e,i} = \exp(-\sum_{j=1}^{i-1} \sigma_{e,j} \delta_j)$. $\tilde{\mathbf{C}}_L$, $\tilde{\mathbf{C}}_B$ and $\tilde{\mathbf{C}}_D$ are a rendered color of the lighting effects $L$, the static background $B$ and the moving objects $D$ in the main stage, respectively.

## 3.2 LEARNING ILLUMINATION VECTOR

Artificial lights in the main stage emit highly saturated colors, and the light intensity varies according to a physical distance between the light source and its projected surface. In contrast, the hue channel of the light source is invariant to the distance. It means that the hue values are globally similar in the whole scene Zimmer et al. (2011). Due to this property, we utilize HSV color space to represent the radiance field for lighting effects, instead of conventional RGB color space.

For hue values $H$, we formulate $v_h$ as a global and learnable component that contains candidate hue information of illuminations along with the time domain. As shown in Fig. 3 (a), for effectively training $v_h$, we initialize the $v_h$ by taking centroids of $k$-means clustering of the hue values that have different colors between the rehearsal and the main stage scene.

Fig. 3 (b) shows that back propagation across the light effect rendering pipeline updates $v_h$. We compute outputs of the neural field $\psi_L$ for volume rendering of the lighting effects; the density $\sigma_L$, saturation channel $S$, value channel $V$ of the radiance and a probability vector $P_H$ whose number of candidates is $k$. $P_H$ is used to calculate the hue channel $H$ corresponding to hue values of the $v_h$ in a time $t$ as:

$$H = \sum_{i=1}^{k} \mathbf{P}_{i,H} v_{i,h}(t), \quad \psi_L : (\mathbf{x}, \mathbf{d}, t) \rightarrow (S, V, \mathbf{P}_H, \sigma_L) \tag{7}$$

$$f_{rgb} : (H, S, V) \rightarrow \mathbf{c}_L, \tag{8}$$

where $v_{i,h}(t)$ is a value sliced over time $t$ and hue index $i$. $\mathbf{P}_{i,H}$ is the probability corresponding to $v_{i,h}(t)$, and $f_{rgb}$ is a function for HSV to RGB channels. In total, we can render light effects using $\sigma_L$ and $\mathbf{c}_L$ converted from the HSV values.

## 3.3 REGULARIZATION FOR ILLUMINATIONS ON DYNAMIC OBJECTS

As done in the baseline Wu et al. (2022), decoupling dynamic and static components can be achieved with two separate neural radiance fields. However, it is infeasible to extract dynamic illuminations because the color changes of the scene do not depend on the motions only, especially for dynamic objects by their time-varying surface normals. We address this problem by leveraging the illumination vector $v_h$ as a prior to identifying dominant $k$ hue values from colors of rays scattered by the light sources. As shown in Fig. 3 (b), $v_h$ facilitates to account for the illuminations on dynamic objects. If the hue values of pixels belonging to moving objects are similar to $v_h$, we can mark the pixels as being entangled with the ray colors. In consideration of this observation, we design a regularization $\mathcal{L}_h$ which eliminates the dynamic parts of objects affected by the lighting effects as follows:

$$\mathcal{L}_h(\mathbf{r}, t) = w_D(\mathbf{r}, t) \, \mathbf{1}_h(\mathbf{r}, t) \tag{9}$$

$$\text{s.t.} \quad \mathbf{1}_h(\mathbf{r}, t) = \begin{cases} 1, & \text{if } \min_{i \leq k}(\|v_{i,h}(t) - f_h(\tilde{\mathbf{C}}_D(\mathbf{r}, t))\|) < \theta \\ 0, & \text{otherwise,} \end{cases} \tag{10}$$

where $\|\cdot\|$ is the standard Euclidean 1-norm, $w_D(\mathbf{r}, t) = \sum_{i=1}^{N} T_{main,i}\rho_{D,i}\alpha_{D,i}$ is the sum of weights for rendering the color of moving objects $D$, $\theta$ is a threshold for an indicator function $\mathbf{1}_h$ and empirically set to $60\,^\circ$. $f_H$ is a function for RGB to hue value. This regularization is only applied to regions of moving objects.

## 3.4 REGULARIZATION FOR ILLUMINATIONS ON STATIC OBJECTS

Apart from the illumination vector $v_h$ for dynamic objects, illuminations on static objects should be considered. The previous works Zhang et al. (2023); Li et al. (2021) focus on decoupling dynamic components from static objects. However, priors used in the previous works (e.g. depth, optical flow and segmentation mask) may not work to decouple the dynamic lighting effects because the lighting colors are highly entangled in pixels of subjects. Instead, we introduce a rehearsal prior taken at the rehearsal stage, which is an easily pre-captured scene before turning on dynamic lighting effects and provides information for scene geometry and subjects' original colors. The rehearsal stage video shows the content of dynamic objects which has similar motion with the main stage, without lighting effects.

We train our radiance fields representing the static objects $\psi_B$ with the rehearsal prior based on a loss $\mathcal{L}_{reh}$ as below:

$$\mathcal{L}_{reh}(\mathbf{r}, t) = \|\mathbf{C}_{main}(\mathbf{r}, t) - \mathbf{C}_{reh}(\mathbf{r}, t) - \tilde{\mathbf{C}}_L(\mathbf{r}, t)\|_2^2 + \|\mathbf{C}_{reh}(\mathbf{r}, t) - \tilde{\mathbf{C}}_B(\mathbf{r})\|_2^2, \tag{11}$$

where $\mathbf{C}_{main}(\mathbf{r}, t)$ and $\mathbf{C}_{reh}(\mathbf{r}, t)$ refer the true color of the camera ray $\mathbf{r}$ at time $t$ in the main stage and the rehearsal stage, respectively.

The first term of $\mathcal{L}_{reh}$ means that we supervise illuminations from lighting effects on roughly disentangled static objects. It enables to decouple the illuminations from the main stage video, even with misaligned pixels between the rehearsal and the main stage video. The second term in $\mathcal{L}_{reh}$ forces the static neural field $\psi_B$ to be trained without the illuminations. Even though there are dynamic objects that are not decoupled with static objects on the rehearsal stage video, our regularization effectively contributes to decoupling light effects and static objects. We note that this loss is only applied to the background regions as well.

| | Light suppression | K-Planes | Semantic K-Planes | D$^2$NeRF | **Ours** |
|---|---|---|---|---|---|
| Stage 1 | 8.62 / 0.247 | 9.07 / 0.493 | 15.86 / 0.543 | 12.33 / 0.740 | **30.04 / 0.889** |
| Stage 2 | 10.73 / 0.290 | 9.19 / 0.427 | 21.03 / 0.702 | 11.95 / 0.695 | **29.48 / 0.840** |
| Stage 3 | 10.86 / 0.216 | 8.81 / 0.420 | 12.06 / 0.205 | 16.57 / **0.865** | **30.13** / 0.816 |
| Stage 4 | 13.75 / 0.274 | 14.98 / 0.601 | 11.76 / 0.292 | 19.59 / **0.886** | **31.21** / 0.856 |
| Mean | 10.99 / 0.257 | 10.51 / 0.485 | 15.18 / 0.436 | 15.11 / 0.797 | **30.22 / 0.850** |

Table 1: Quantitative evaluation with the state-of-the-art methods (PSNR/SSIM).

## 4 EXPERIMENTS

### 4.1 IMPLEMENTATION DETAILS

The neural fields ($\psi_B$, $\psi_D$ and $\psi_L$) of our ReHeaRF are implemented with a hybrid model of K-Planes Fridovich-Keil et al. (2023). Note that the neural fields can be replaced by other dynamic NeRF models. A training procedure of ReHeaRF requires a day for 300k iterations with 8,192 batch sizes on a single RTX 3090 GPU. We train our ReHeaRF with the following loss $\mathcal{L}$ :

$$\mathcal{L}(\mathbf{r}, t) = \mathcal{L}_{recon}(\mathbf{r}, t) + \lambda_m \mathcal{L}_m(\mathbf{r}, t) + \lambda_{reh} \mathcal{L}_{reh}(\mathbf{r}, t) + \lambda_h \mathcal{L}_h(\mathbf{r}, t) \tag{12}$$

$$\text{s.t.} \quad \mathcal{L}_{recon}(\mathbf{r}, t) = \|\mathbf{C}_{main}(\mathbf{r}, t) - \tilde{\mathbf{C}}_{main}(\mathbf{r}, t)\|_2^2, \tag{13}$$

$$\text{and} \quad \mathcal{L}_m(\mathbf{r}, t) = \mathbf{M}_B \sum_{i=1}^{N} T_{main,i} \rho_{D,i} \alpha_{D,i} + \mathbf{M}_D \sum_{i=1}^{N} T_{main,i} \rho_{B,i} \alpha_{B,i}, \tag{14}$$

where $M_\Omega$ is a mask which indicates a valid region of a component $\Omega \in [B, D]$. Using an off-the-shelf interactive segmentation Cheng et al. (2021) in the offline phase, we identify the valid region $M_\Omega$ in both the rehearsal and main videos of our dataset. $\mathcal{L}_{recon}$ is a reconstruction loss and $\mathcal{L}_m$ is a regularizer which removes regions where the components are disentangled. In our implementation, $\lambda_m$, $\lambda_{reh}$ and $\lambda_h$ are empirically set to 0.01, 0.5 and 0.01, respectively. In the experiments, we set the number of clustering of $k$-means algorithm to 5 because of the number of light sources in the main stages.

### 4.2 DATASETS

Since public datasets for dynamic NeRFs Li et al. (2022b); Broxton et al. (2020) are limited to scenes captured under stable illuminations, we construct a video dataset for dance stages taken under both stable and dynamic colored lighting environments. Following the problem setting, our dataset contains pre-captured videos under the stable lighting conditions before turning on the dynamic illuminations as well as camera parameters obtained from COLMAP Schönberger & Frahm (2016).

We build a dataset consisting of synchronized multi-view videos whose spatial and temporal resolutions are 2704×1520 and 120 FPS, respectively. For this, we use an array of 20 GoPro HERO cameras and take 10 video clips whose running time is 2.5 seconds. In the experiment, we downsample the spatial resolution by a factor of 2 and the frame rate to 30 FPS, similar to the previous work Li et al. (2022b).

Note that we do not directly compare our ReHeaRF with other methods on this dataset because, to the best of our knowledge, our work is the first attempt to decouple dynamic lighting effects from dynamic objects in NeRFs. Instead, we evaluate the ability to disentangle lighting effects when static parts are only rendered.

### 4.3 COMPARISON WITH STATE-OF-THE-ART METHODS

We demonstrate the capability of our ReHeaRF to decouple lighting effects from subjects. As mentioned earlier, our work is the first attempt to handle dynamic illuminations in NeRF, and we cannot carry out an apple-to-apple comparison with publicly available neural rendering methods. Therefore, we could make only a limited evaluation of dynamic lighting effects. To do this, we manually mask out moving objects in scenes

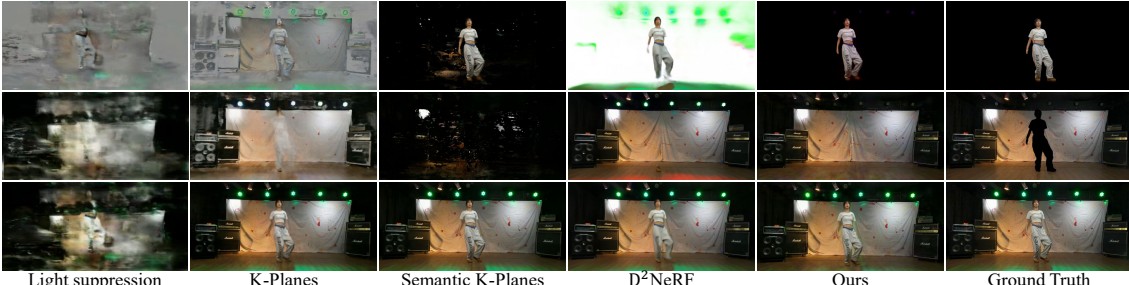

Light suppression     K-Planes     Semantic K-Planes     D²NeRF     Ours     Ground Truth

Figure 4: Qualitative comparison ours with the state-of-the-art methods. The ground-truth scenes are captured in the rehearsal stages. Note that we only compare the static background region ($2^{nd}$ row) because moving objects in the rehearsal scene are considered as outliers in the evaluation on the disentanglement.

| | Mean | | | Mean | |
|---|---|---|---|---|---|
| | PSNR | SSIM | (15 frames) | PSNR | SSIM |
| w/o hue init. | 29.12 | 0.822 | +1.0s | 28.65 | 0.793 |
| w/o $\mathcal{L}_{reh}$ | 9.36 | 0.481 | +2.0s | 28.53 | 0.792 |
| w/o $\mathcal{L}_h$ | 29.57 | 0.818 | w/o motion | 28.10 | 0.762 |
| w/o $\rho$ | 29.12 | 0.805 | | | |
| **Ours** | **29.73** | **0.825** | **Ours** | **28.66** | **0.801** |

Table 2: Ablation study. Note that the results are on static parts only and mearsured for 10 scenes of our dataset. (Left) The effectiveness of each component in our framework; (Right) Study on mis-syncronization between the main and the rehearsal stages.

and evaluate a photo-consistency between the remaining regions of the rehearsal stage and its corresponding regions of rendered static parts using PSNR and SSIM Wang et al. (2004) metrics. In this experiment, we compare our ReHeaRF with D²NeRF, which is our baseline model, and following state-of-the-art methods: (1) K-Planes Fridovich-Keil et al. (2023): It consists of six planes to represent dynamic motions. By fixing temporal planes in inference time, it can render static parts only.

(2) Semantic K-Planes: Inspired by SemanticNeRF Zhi et al. (2021), we implement SemanticNeRF by adding semantic logits, classes for static background, moving objects, and lighting effects, to K-Planes. For the comparison with Semantic K-Planes, we use masks ($M_\Omega$) and labels associated with lighting effects. However, labeling lighting effects manually can be challenging. We thus annotate positions of lighting effects in areas where ISG weights Shuai et al. (2022) are high and dynamic objects are not present.

(3) Light suppression Jin et al. (2022): Additionally, we use light suppression as a comparison method. We train K-Planes on videos where the light suppression method is applied.

As shown in Table 1, we report the quantitative evaluation result. Since the lighting effects have a significant impact on PSNR values, if they are incorrectly decoupled, the predicted colors of objects differ from their original colors. Both K-Planes and D²NeRF, trained using photo-realistic loss under dynamic lighting, often predict the average colors of objects over the entire time. On the other hand, as expected, our ReHeaRF consistently achieves the best performance over the comparison methods.

In Fig. 4, we display an example of the quantitative results in Table 1. Fig. 4 qualitatively shows the huge performance differences between ours and the competitive methods as well. First, the light suppression method never works in our dataset because it disrupts the view consistency among multiple cameras. Both K-Planes and D²NeRF also suffer from extracting moving objects. We observe that solid objects are treated as dynamic parts due to the varying light colors over time. Although Semantic K-Planes utilizes semantic information to decouple dynamic objects, the lighting effects still exist on the surfaces of the objects. As a result, the varying colored regions by the lighting effects disappear. In contrast to, our ReHeaRF successfully disentangles these factors while correctly representing each neural field.

Lastly, we show two interesting applications of our work to video manipulations and confocal florescence microscopy. Please check them in appendix A.1 and A.2 as well as the submitted video.

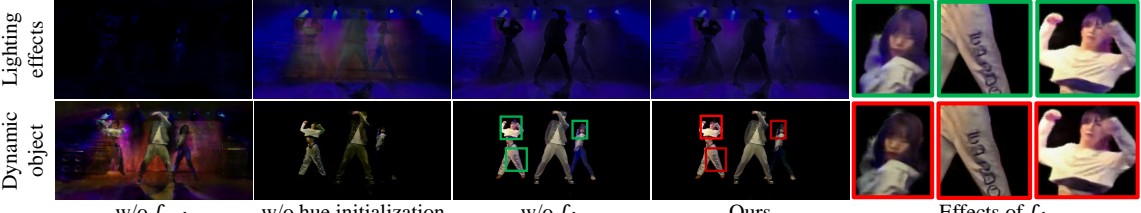

Figure 5: Visualizations of the results on ablation study. We can see that each component of our ReHeaRF contributes to the visually pleasing looks. Particularly, when we impose $\mathcal{L}_h$, ReHeaRF shows distinctively realistic subjects which is decoupled with scene lights.

### 4.4 ABLATION STUDY

In Table 2 (Left) and Fig. 5, we demonstrate the effectiveness of the rehearsal prior $\mathcal{L}_{reh}$, $\mathcal{L}_h$, the hue initialization and our novel volume rendering ($\rho$ in Eq. (4)). Without $\mathcal{L}_{reh}$, the dynamic objects and the static part are trained independently, which learns to render dynamic and static parts at once. In addition, if we remove similar color regions between objects and the illumination vector using $\mathcal{L}_h$, some parts of dynamic objects may be missed in the neural field for dynamic objects, and the missed parts are evident when rendering lighting effects. In Table 2 (Left), we observe that $\mathcal{L}_h$ has a positive impact on decoupling lighting effects on static parts of scenes even though it is related to colors on dynamic objects. Because the illumination vector accounts for all lighting effects in ReHeaRF framework, the vector representation can be enhanced with $\mathcal{L}_h$, which yields better decoupling results on static regions. The hue initialization allows the illumination vector to effectively learn global information. Therefore, without the hue initialization, the $\mathcal{L}_h$ could mistakenly remove dynamic object parts which are not affected by lighting effects. Lastly, $\rho$ in Eq. (4) is also effective to disentangle lighting effects. The reason is why $\rho$ breaks the common assumption of NeRF that two objects cannot be located in the same position.

In addition, we show that our ReHeaRF does not suffer from mis-synchronization between rehearsal and main stages in Table 2 (Right). We intentionally make mis-synced videos by shifting the time axis of the rehearsal stages by 1.0 second and 2.0 seconds. We conduct this experiment using 0.5-second (15 frames) video clips because its time duration is 2.5 seconds and the video used will be shifted by 2.0 at most in our dataset. Additionally, we only use the first frame of the rehearsal stage video which means no motion information is used. In this study, we do not observe any significant performance drop due to mis-synchronization and no motion information. The reason for this is that our ReHeaRF is designed to train only the geometry and appearance from the rehearsal stage through $\mathcal{L}_{reh}$.

## 5 CONCLUSION

In this paper, we present a novel approach that trains and renders neural fields for dynamic scenes captured under drastically changing illuminations. The proposed method decouples dynamic lighting effects from static/moving objects. For this, our key idea is to devise the rehearsal prior, which can be easily taken before turning on lights for the main stage. In addition, we leverage the light color-adaptive vectors and the semantic-aware regulerizations to jointly optimize neural fields for static/dynamic objects and varying illuminations. We demonstrate the effectiveness of the proposed method by showing impressive results on novel view synthesis under dynamic illuminations.

**Limitation** & **Future directions** Directions for improvement exist. First, since our work focuses on dynamic scene representations for short clips, efficient training schemes are needed for long sequence videos. In addition, we assume that the camera are static during taking videos. For more generality of this work, solutions to reconstructing neural fields using moving camera setups can be one of interesting future works.

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

# A APPENDIX

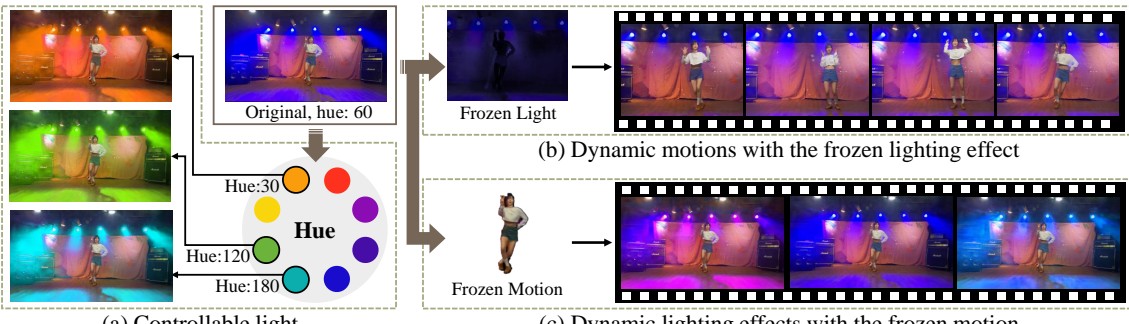

Figure 6: Applications of ReHeaRF to video manipulation. (a) We can change the light colors after recording the video. (b) Our ReHeaRF produces the natural-looking dance video under artificially stopping the light effects. (c) We show a video with the frozen motion, but the light sources still work. Please check our supplementary video.

## A.1 APPLICATION1: VIDEO MANIPULATION

Accurate disentanglement of neural fields for dynamic lighting effects and scene objects in the time domain can facilitate many applications. As examples, we show video editing applications such as the slow motion and time stop effect (a.k.a bullet-time) by manipulating the temporal flow of the decoupled neural fields.

A controllable light effect that tunes light colors after taking the video is one of the applications of our ReHeaRF. Thanks to the accurate neural representation for dynamic lights, we can artificially direct the various light color effects in Fig. 6(a). Another interesting application is to freeze the subject's motion and light colors. As shown in Fig. 6(b) and (c), since the illumination neural fields are supervised independently for the motion and geometry of the subject, we can easily produce the videos that show natural-looking motions in artificially fixed light conditions and vice versa.

## A.2 APPLICATION2: CONFOCAL FLUORESCENCE MICROSCOPY

| | Confocal fluorescence microscope | ReHeaRF |
|---|---|---|
| Illumination | various wavelength LEDs/Lasers | dynamic illuminations at a concert hall |
| Sensor | multiple detectors | multiple cameras |
| Subject | live cells and tissues | performers |
| Output | 3D volume rendering of live cells and tissues | radiance fields for components of a scene |

Table 3: Similarity of input/output configurations between confocal fluorescence microscopy and ReHeaRF.

We demonstrate that our ReHeaRF can also be extended to other domains. As an example, we show a potential application to a confocal fluorescence microscope. Before describing the feasible scenario of our framework, we would like to introduce a concept of the confocal fluorescence microscope.

The confocal fluorescence microscope is an optical microscope to scan cells and tissues and to be used for studying the properties of them. As the name suggests, the microscopes use the fluorescence to generate an image with much higher intensity light sources which excite fluorescent species in a sample of interest. And, the confocal fluorescence microscope is devised to take higher resolutions and contrast images than that of the fluorescence microscope by blocking out-of-focus lights in image formulations. Capturing multiple 2D images

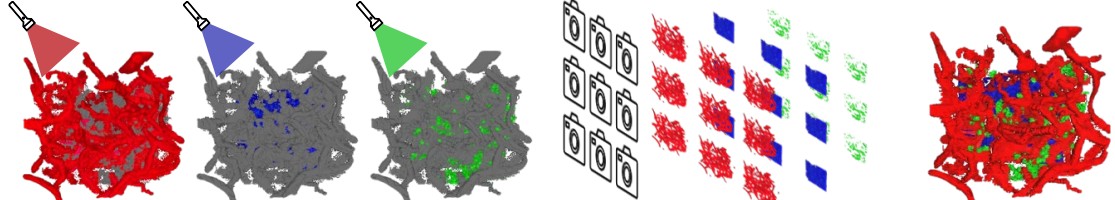

(a) Activation depending on the wavelengths  (b) Multi-view observation  (c) Volume Rendering

Figure 7: Applying ReHeaRF framework to confocal fluorescence microscopy. (a) Each sample type exhibits different activations according to the wavelengths of exposed lights. (b) Through multi-view observation such as light field microscopy, we can generate multi view images which can be used to train ReHeaRF. (c) We can render the 3D volume of the subject. Please check our supplementary video.

at either different depths Wang et al. (2021a) or various viewpoints (i.e. light field microscopy Browning et al. (2022)) in a sample enables 3D reconstruction and volume rendering of its structure. In particular, it becomes essential to investigate spatial arrangement of live cells and tissues with high precision in sciences, which is useful for assigning the localization to specific cellular compartments or finding out the relationships between them Jonkman et al. (2020).

In total, we claim that the system configuration, inputs and outputs of the confocal fluorescence microscopes are very similar with those of our Rehearsal NeRF as described in Table 3:

Following this scenario of the confocal fluorescence microscope, we provide an example to demonstrate the potential application. In Hontani et al. (2021), a set of images for a mouse brain captured using a fluorescence microscopy is available. Using the images, we first make a composite image and then project it onto a regular grid for the multiview image acquisition, which becomes the input of our ReheaRF. By training ReHeaRF with these images, we can synthesize novel views of specific cells or entire of cells. As shown in Fig. 7, our ReheaRF achieves the well-reconstructed spatial arrangement of the mouse brain. In the submitted video demo, we present rendering results that are synthesized by our ReHeaRF; please refer it. If this framework were extended to a temporal dataset consisting of flourescence microscope images, it could lead to the discovery of many interesting properties of cells and tissues by expanding into 4D domain (3D + time).

## A.3 DIVERSITY OF OUR DATASET

| Dataset | # of scenes | # of dynamic subjects | # of backgrounds |
|---|---|---|---|
| Nerfies Validation Rig Park et al. (2021a) | 4 | 3 | 3 |
| Neural 3D video Li et al. (2022b) | 6 | 1 | 2 |
| Meet Room Li et al. (2022a) | 3 | 3 | 1 |
| Ours | 10 | 4 | 3 |

Table 4: Statistic of public datasets for NeRFs with dynamic motion and our dataset

Since the advent of NeRF, new datasets have been taken whenever new problems are defined. We collect a new dataset with dynamic scenes with changing illumination, a unique situation that has not been addressed before. Our dataset ensures the diversity with respect to the number of dancers, choreography, costumes, and backgrounds, compared to the public datasets for NeRFs, as reported in Table 4. Furthermore, our dataset contains songs corresponding to the dances, which can be useful for multi-modal learning applications such as motion generation based on music.

| | Stage 1 | | Stage 2 | | Stage 3 | | Stage 4 | | Stage 5 | |
|---|---|---|---|---|---|---|---|---|---|---|
| | PSNR | SSIM | PSNR | SSIM | PSNR | SSIM | PSNR | SSIM | PSNR | SSIM |
| w/o hue init. | 26.55 | 0.798 | 30.21 | **0.840** | **30.47** | **0.809** | 29.53 | 0.823 | 29.18 | 0.811 |
| w/o $\mathcal{L}_{reh}$ | 6.61 | 0.446 | 7.20 | 0.477 | 13.40 | 0.558 | 13.51 | 0.537 | 8.32 | 0.408 |
| w/o $\mathcal{L}_h$ | 28.31 | 0.811 | **30.78** | 0.833 | 29.65 | 0.778 | **31.21** | **0.846** | 29.83 | 0.813 |
| w/o $\rho$ | 29.14 | 0.822 | 29.19 | 0.825 | 29.09 | 0.801 | 29.69 | 0.735 | 29.10 | 0.794 |
| **Ours** | **30.00** | **0.832** | 29.46 | 0.828 | 30.09 | 0.802 | 31.18 | 0.831 | **30.08** | **0.819** |
| | Stage 6 | | Stage 7 | | Stage 8 | | Stage 9 | | Mean | |
| | PSNR | SSIM | PSNR | SSIM | PSNR | SSIM | PSNR | SSIM | PSNR | SSIM |
| w/o hue init. | 29.19 | 0.861 | 29.24 | 0.805 | 29.20 | **0.779** | 28.55 | 0.869 | 29.12 | 0.822 |
| w/o $\mathcal{L}_{reh}$ | 7.36 | 0.467 | 7.26 | 0.434 | 8.29 | 0.505 | 12.26 | 0.496 | 9.36 | 0.481 |
| w/o $\mathcal{L}_h$ | 28.00 | 0.840 | 29.36 | 0.794 | **29.64** | 0.762 | 29.38 | 0.885 | 29.57 | 0.818 |
| w/o $\rho$ | 28.37 | 0.847 | **29.90** | 0.802 | 28.82 | 0.740 | 28.82 | 0.883 | 29.12 | 0.805 |
| **Ours** | **29.84** | **0.867** | 29.47 | **0.811** | 28.83 | 0.759 | 28.67 | 0.872 | **29.73** | **0.825** |

Table 5: Ablation experiments of each component in our framework.

| | Stage 1 | | Stage 2 | | Stage 3 | | Stage 4 | | Stage 5 | |
|---|---|---|---|---|---|---|---|---|---|---|
| | PSNR | SSIM | PSNR | SSIM | PSNR | SSIM | PSNR | SSIM | PSNR | SSIM |
| $+1.0s$ | 27.88 | 0.781 | 29.18 | 0.808 | 29.57 | 0.752 | **30.14** | **0.813** | 27.78 | 0.773 |
| $+2.0s$ | 27.83 | 0.781 | **29.61** | 0.806 | 29.72 | 0.778 | 29.06 | 0.743 | **29.10** | **0.794** |
| w/o motion | 27.36 | 0.766 | 29.17 | 0.787 | 28.26 | 0.704 | 29.62 | 0.780 | 27.10 | 0.754 |
| **Ours (15 frames)** | **28.76** | **0.793** | 29.36 | **0.809** | 30.94 | **0.817** | 29.62 | 0.777 | 28.12 | 0.782 |
| | Stage 6 | | Stage 7 | | Stage 8 | | Stage 9 | | Mean | |
| | PSNR | SSIM | PSNR | SSIM | PSNR | SSIM | PSNR | SSIM | PSNR | SSIM |
| $+1.0s$ | **28.05** | **0.831** | 28.26 | 0.756 | 28.95 | 0.752 | 28.01 | 0.875 | 28.65 | 0.793 |
| $+2.0s$ | 26.79 | 0.820 | 28.15 | 0.758 | **29.77** | **0.780** | 26.80 | 0.869 | 28.53 | 0.792 |
| w/o motion | 27.10 | 0.755 | 26.99 | **0.815** | 28.38 | 0.773 | **28.72** | 0.726 | 28.10 | 0.762 |
| **Ours (15 frames)** | 27.57 | 0.825 | 27.41 | 0.764 | 29.41 | 0.760 | 28.70 | **0.883** | **28.66** | **0.801** |

Table 6: Ablation study on mis-synchronization of motions between rehearsal and main stages. Note that we only utilize 15 frames (0.5 seconds) on this experiment.

## A.4 ABLATION STUDY: DETAILS OF ABLATION STUDY IN SEC. 4.4

Since we only report the mean PSNR and SSIM of 9 scenes in Table 2, we provide details of the ablation study for each scene in Table 5 and Table 6.

## A.5 ABLATION STUDY: HSV COLOR SPACE

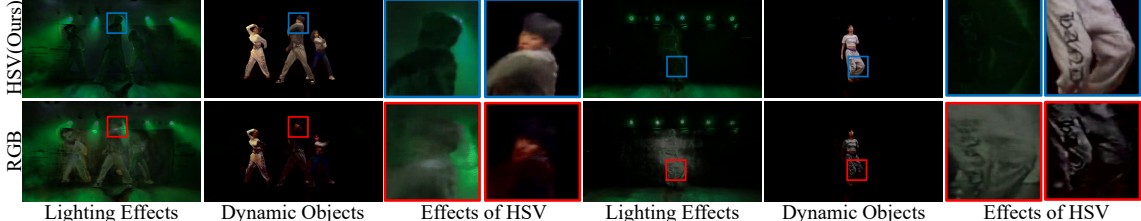

| Lighting Effects | Dynamic Objects | Effects of HSV | Lighting Effects | Dynamic Objects | Effects of HSV |

Figure 8: Ablation experiments on HSV color space.

Since the hue channel of the light source is invariant to the distance, we select HSV color space to represent the illumination vector. When we utilize RGB color space to represent the illumination vector, as shown in Fig. 8, we observe that the illumination vector suffers from distinguishing lighting effects and dynamic objects. The reason is why the RGB color space contains hue, saturation and value altogether.

## A.6 CASE1: MIS-ALIGNMENT BETWEEN MAIN AND REHEARSAL STAGES

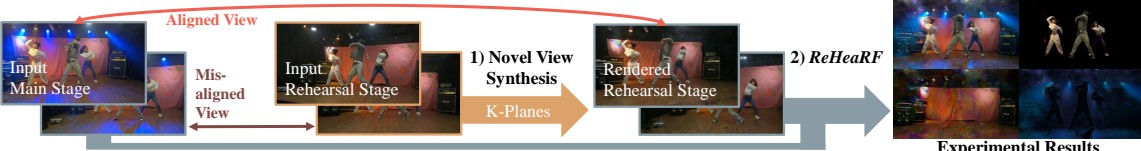

Figure 9: Handling a large misalignment of cameras between main and rehearsal stage.

Our work assumes that the location of the camera array is fixed during taking both the main and rehearsal stage videos like real performance scenarios. However, when a large misalignment exists, we can address the issue using a multi-view video-based view synthesis method. For this, we conduct an experiment with different camera poses between the rehearsal and live stages. In this experiment, we initially train K-Planes using rehearsal videos. Subsequently, we utilize the trained K-Planes to synthesize rehearsal videos at the camera poses of the live videos. The result is displayed in the Fig. 9.

## A.7 CASE2: HANDLING COMPLEX LIGHTING EFFECTS.

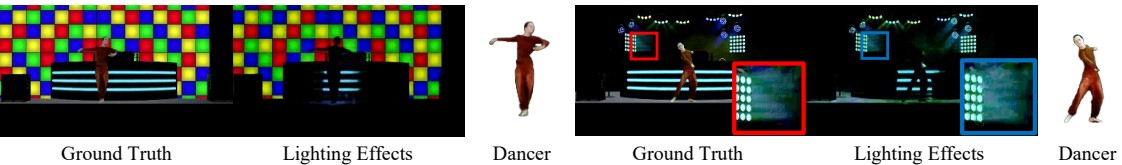

Figure 10: Experimental results on complex lighting effects.

Intuitively, very complex lightings, such as a bulky LED array in the background, may result in performance drops of the proposed method. However, we experimentally demonstrate that our hue initialization and illumination vectors in a time domain are beneficial as they provide sufficient information about intense dynamic lightings. For this, we conduct an experiment with a synthetic dataset simulating a scenario that a dancer is captured under a bulky back-sided LED array. Note that we assume that there is no saturation on scenes. We bought commercial Unity assets for both the dancer and stage with the bulky LED array. Our ReHeaRF successfully decouples the lighting effects by adjusting the hyper-parameter $k$, which represents the number of clusters in the k-means algorithm, which is shown in Fig. 10

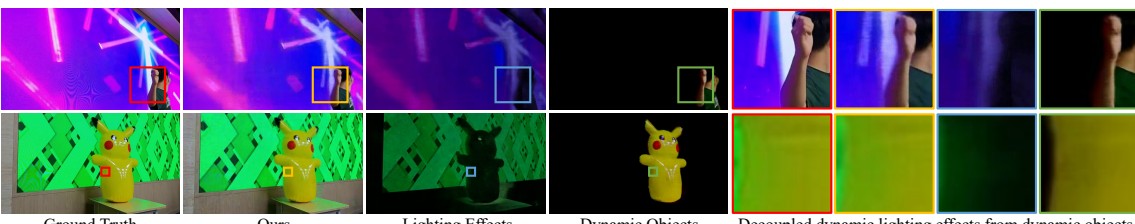

Figure 11: Experimental results when the subjects are captured in front of the bulky LED screen. Please check our supplementary video.

We also carry out the real-world experiment in scenes in front of a large LED screen, which is commonly used in concerts as a popular light source. The resolution of the large LED screen we used is $3360 \times 1620$ (5,443,200 light sources). As displayed in Fig. 11, ReHeaRF effectively separates dynamic lighting effects from objects in this scene.

### A.8 CASE3: THE COLOR OF THE DANCER'S CLOTHING IS SIMILAR TO THE SPOTLIGHTS.

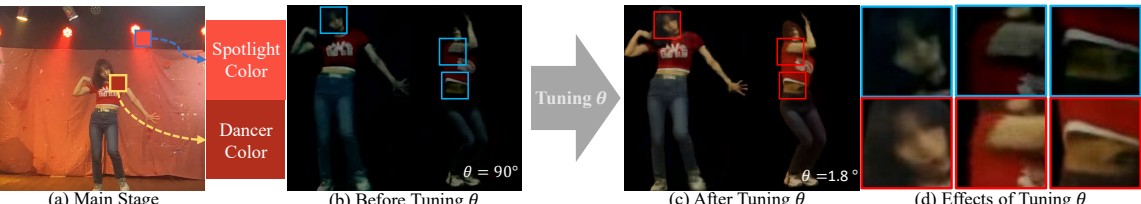

(a) Main Stage    (b) Before Tuning $\theta$    (c) After Tuning $\theta$    (d) Effects of Tuning $\theta$

Figure 12: Although the color of the dancer's clothing is similar to the spotlight, ReHeaRF is still working well by tuning $\theta$ of $L_h$.

In Fig. 5, we demonstrate that $L_h$ effectively decouples lighting effects from dynamic objects. However, because $L_h$ relies on color differences between dynamic objects and lighting effects, it may encounter difficulties when the colors of lighting effects and dynamic objects are similar. But, in our framework, adjusting both the weight parameter $\lambda_h$ and the threshold $\theta$ which distinguishes lighting effects within the radiance field of dynamic objects, can effectively address this issue. As an example, in Fig. 12, adjusting $\theta$ to 1.8 degrees allows us to effectively decouple dynamic objects from lighting effects.

### A.9 CASE4: IF THE COMPARISON METHODS USE THE REHEARSAL PRIOR.

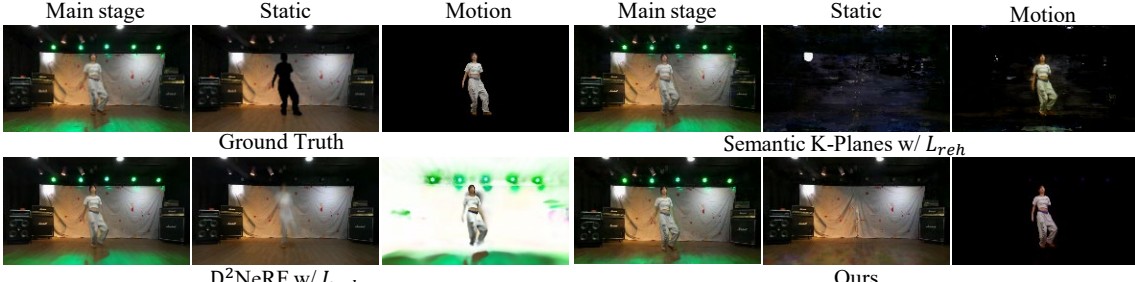

Figure 13: Qualitative evaluations with the comparison methods with the rehearsal prior $L_{reh}$.

In this paper, the comparison methods did not use rehearsal videos because we want to directly compare ours with them as it is. For fair comparisons, we train the static parts for D$^2$NeRF and Semantic K-Planes with rehearsal priors, and compare their rendering qualities. As shown in Fig. 13, with the rehearsal priors, D$^2$NeRF still suffers from extracting moving objects, and Semantic K-Planes fails to decouple static object.

