# OpenReview forum: "Rehearsal NeRF: Disentangling Dynamic Illuminations in Neural Radiance Fields"
_ICLR.cc/2024/Conference — ICLR 2024 Conference Withdrawn Submission_

### Official Review · Reviewer_jMZ5 · 2023-10-30

**Soundness:** 4 excellent
**Presentation:** 3 good
**Contribution:** 4 excellent
**Rating:** 6
**Confidence:** 4

**Summary:**

This paper considers a new challenging task of decoupling dynamic and static objects from a dynamic scene with time-varying illuminations. To this end, this work proposes a ReHearsal prior-based Radiance Fields (ReHeaRF) by taking both rehearsal and main stage videos as input. With the rehearsal video prior, a regularization loss is designed for robust supervision to decouple static objects and illuminations. A learnable illumination vector is also introduced to render the dynamically changing illuminations via a neural illumination field. Extensive experiments on a newly built video dataset demonstrate the effectiveness of the proposed ReHeaRF by showing impressive results on novel view synthesis under dynamic illuminations.

**Strengths:**

+ This paper focuses on an important, understudied problem.
+ By utilizing an additional rehearsal video as a prior, the technical route of the proposed method seems reasonable.
+ The quantitative results in Table 1 show that the proposed method outperforms the other competing methods overall, especially the PSNR metric.

**Weaknesses:**

- Visually, the qualitative results provided in Figure 6 and the appendix video show that the lighting effects with the frozen motion are not particularly obvious as the color of the background light source changes.
- The presentation of this paper could be improved. For example,
The arrow between the \psi_{D} and the c_{D} is missing in Figure 2;
Is the viewing direction three-dimensional in Eq. 2?
a illumination neural field -> an illumination neural field.

**Questions:**

Here are some questions that need to be addressed in the rebuttal.
-	As mentioned in the weakness part, the lighting effects with the frozen motion are not particularly obvious. I noticed that the light sources on the stage are almost placed behind the dancers, which results in a limited surface area where the light source directly illuminates the human body. I wonder what the performance of the proposed method would be like if the light sources were placed in front of the dancers. Would the lighting effects be more obvious?
-	In Figure 3, it can be observed that the main stage mainly consists of five light sources. K of the top-k hue is also set to 5. How is k determined? What is the effect of different k on the proposed method?

---

### Official Review · Reviewer_cgWH · 2023-10-31

**Soundness:** 2 fair
**Presentation:** 2 fair
**Contribution:** 2 fair
**Rating:** 5
**Confidence:** 4

**Summary:**

The paper posits a novel problem and its corresponding solution. It harnesses the NeRF methodology to disentangle light effect from dynamic performance videos. For such purposes, a rehearsal video is recorded to provide two priors. The first is the observation of the static stage without light effect, while the second is potential light effect by subtracting rehearsal video pixel from main performance video pixel. The problem is novel and practical and the solution looks effective.

**Strengths:**

1. The proposed method is novel and practical. In many performance the stage lighting is dynamic however few paper aims to recover 3D representation from these videos. The proposed problem could be helpful to solve the problem.
2. Some specific techniques are proposed to facilitate the method. For example, the rehearsal video is subtracted from main stage video to provide potential light effects, and regularizations are adopted to force the dynamic object influenced as little as possible by stage light. From the ablation study the proposed techniques are effective to disentangle static objects, dynamic objects and dynamic light.

**Weaknesses:**

1. The problem is novel however the solution may lack rationality. The static objects, dynamic objects, and light effect are implemented with NeRF and added together. It seems the light effect NeRF acts as a dynamic fog wraped around static/dynamic objects, which may be unreasonable. Ideally the lighted pixel should be produced by rendering static/dynamic objects slightly influenced by light effect NeRF, but not by added them together.
2. The rehearsal video provides Hue prior and static objects prior. It seems that the performance makes no use and the prior can be provided by simply capturing the empty stage.
3. The rehearsal video plays the vital role in the novelty however it is not well explained. It only appears in Sec. 3.4 as regularization on static objects and I think it can be replaced by capturing empty stage. I think the writing could focus on how to leverage rehearsal video to show its importance.

**Questions:**

1. I need explaination of the rationality of adding static/dynamic objects and light effect together.
2. Can the rehearsal video be replaced by capturing static stage?

---

### Official Review · Reviewer_GjAK · 2023-11-03

**Soundness:** 3 good
**Presentation:** 3 good
**Contribution:** 2 fair
**Rating:** 3
**Confidence:** 4

**Summary:**

The paper introduces ReHeaRF, a method designed to address the challenge of rendering and reconstructing neural radiance fields in the context of dynamic illumination changes. It utilizes scenes captured in consistent lighting to maintain geometric stability and applies a learnable vector to separate light colors from scene radiance. The approach includes a proposed regularizer to distinguish dynamic objects within the scene. Comparative results and supplementary materials, including source codes and a video demo, are provided to illustrate the method's application.

**Strengths:**

- The paper is well-structured, facilitating comprehension, and commendably provides supplementary materials such as a video demonstration and source code. The effort invested in capturing the dataset is also evident and contributes to the depth of the research.

- The methodology employed by ReHeaRF to disentangle dynamic illuminations from objects in a scene employs a learnable illumination vector. This feature is designed with the intent to enhance the precision of rendering in various lighting environments.

**Weaknesses:**

- The paper could benefit from a clearer articulation of the motivation behind using two input stages. It remains unclear why scenes captured during the rehearsal stage, which lack dynamic lighting, are insufficient for the desired outcome, necessitating the use of additional inputs with dynamic lighting.

- The assertions regarding the necessity of optimizing dynamic illuminations for effective separation of dynamic and static objects warrant a more rigorous comparative analysis with state-of-the-art (SOTA) masking techniques to substantiate this claim.

- The quality of the results concerning light decomposition on subjects suggests there is room for improvement. Visible light effects on individuals within the scene indicate that the method may not fully isolate the illumination effects as intended.

- When it comes to experimental validation, the comparison seems limited to dynamic NeRF methods like K-Planes and D2NeRF, which primarily address dynamic view synthesis. A broader evaluation, including a qualitative and quantitative comparison with neural inverse rendering methods like NeRD or NeRFactor, would be valuable. Additionally, the omission of citations for some relevant inverse rendering papers should be addressed to provide a comprehensive reference framework:
```
Iron: Inverse rendering by optimizing neural sdfs and materials from photometric images,2022
Physg: Inverse rendering with spherical gaussians for physics-based material editing and relighting,2023
Nerfactor: Neural factorization of shape and reflectance under an unknown illumination.2023
Relit-NeuLF: Efficient Relighting and Novel View Synthesis via Neural 4D Light Field, 2023
TensoIR: Tensorial Inverse Rendering,2023
Modeling indirect illumination for inverse rendering,2022
```
- More details on the hardware setup are needed. For example, the paper should mention which GoPro camera model was used and how the cameras were synchronized.

**Questions:**

While the paper presents a promising approach and valuable supplementary materials, I recommend a weak rejection at this stage. The innovative concept and the effort to provide a clear, practical understanding of the system are commendable. However, for a stronger resubmission, clarifying the motivation for the dual-input methodology, reinforcing the claims about dynamic illumination optimization with more evidence, and refining the decomposition results to address visible lighting artifacts are suggested. Additionally, elaborating on the hardware setup would greatly enhance the reproducibility and robustness of the work.

---

### Official Review · Reviewer_3545 · 2023-11-06

**Soundness:** 2 fair
**Presentation:** 2 fair
**Contribution:** 2 fair
**Rating:** 3
**Confidence:** 3

**Summary:**

This paper proposes a NeRF-based method for rendering dynamic scenes under varying lighting conditions; it jointly optimizes neural fields for static/moving objects and varying light sources. Specifically, the proposed method assumes scenes such as dancing on a stage, and decouples the effects of static/moving objects and varying lighting conditions by taking the rehearsal prior into consideration; videos taken with similar object’s motion under neutral lighting condition. The performance of the proposed method is compared with those of the state-of-the-art methods on their own dataset captured from 20 static cameras.

**Strengths:**

First of all, this paper tackles a novel and challenging problem of rendering dynamic scenes under varying lighting conditions. Second, the experimental results demonstrate that the proposed method works better than the state-of-the-art methods (Light suppression and K-Planes) as well as the baseline method (D^2NeRF).

**Weaknesses:**

I think that the applicability of the proposed method would be quite limited due to the following two reasons. First, the proposed method requires the rehearsal videos captured with similar object’s motion under neutral lighting condition. Such videos would be available for scenarios such as dancing on a stage, but it is not clear whether they are easily available for other scenarios. Second, judging from the figures in the paper and the videos in the supplementary materials, the proposed method assumes that the light sources are directly visible from cameras and illuminate moving objects from behind. In other words, the estimation of the light source’s colors is easy, and the effects of the light source’s colors on the moving objects are relatively limited. Therefore, it is not clear whether the proposed method is applicable to scenes under frontal light sources located behind cameras. In addition, the presentation of this paper could be improved. In particular, Sections 3.2 and 3.3 are difficult to read, because the definitions of v_h and v_{I,h} are not clear enough.

**Questions:**

I would be happy to receive your feedback to the comments on Weaknesses.